# A rapid access to aliphatic sulfonyl fluorides

Ruting Xu[1], Tianxiao Xu[1], Mingcheng Yang[1], Tianpeng Cao[1] & Saihu Liao ⓘ [1]

The past few years have witnessed a fast-growing research interest on the study of sulfonyl fluorides as reactive probes in chemical biology and molecular pharmacology, which raises an urgent need for the development of effective synthetic methods to expand the toolkit. Herein, we present the invention of a facile and general approach for the synthesis of aliphatic sulfonyl fluorides via visible-light-mediated decarboxylative fluorosulfonylethylation. The method is based on abundant carboxylic acid feed stock, applicable to various alkyl carboxylic acids including primary, secondary, and tertiary acids, and is also suitable for the modification of natural products like amino acids, peptides, as well as drugs, forging a rapid, metal-free approach to build sulfonyl fluoride compound libraries of considerable structural diversity. Further diversification of the $SO_2F$-containing products is also demonstrated, which allows for access to a range of pharmaceutically important motifs such as sultam, sulfonate, and sulfonamide.

[1] Key Laboratory for Molecule Synthesis and Function Discovery (Fujian Province University), College of Chemistry, Fuzhou University, Fuzhou 350116, China. Correspondence and requests for materials should be addressed to S.L. (email: shliao@fzu.edu.cn)

Rapid assembly of compound libraries with high efficiency and diversity is critical to drug discovery. By virtue of the combination of strong thermodynamic driving forces and well-controlled reaction pathways, click reactions afford bond construction in an operationally simple, modular, efficient, and reliable manner. Since the introduction of the conceptual framework by Sharpless and co-workers, click chemistry has gained a great impact on many areas in the last two decades, not only organic synthesis but also biology, medicine, and materials science[1]. Among various reactions identified for click chemistry, Sulfur(VI) Fluoride Exchange (SuFEx) represents the latest and also one of the most powerful click reactions[2–5]. Next to the recent significant progress in synthetic methodology development and polymer preparation[6–11], another intriguing application of SuFEx chemistry could be the use of sulfonyl fluorides in chemical biology and molecular pharmacology as a privileged type of warheads[12–21]. The unique properties observed with sulfonyl fluorides could be ascribed to (i) the right balance of inherent electrophilic reactivity of $S^{VI}$–F and stability under physiological aqueous conditions, and (ii) the proton ($H^+$)-mediated reactivity-switch-on mechanism that makes its reactivity sensitive to the microenvironment of the binding site, and thus allows a site-specific targeting under various chemical and biological contexts[2,12–14]. The successful identification of their highly selective inhibition activity has boosted a fast-growing research interest in recent years to develop enzyme inhibitors or chemical probes based on sulfonyl fluorides[12–26]. However, the development in this area is significantly hampered by the limited availability of sulfonyl fluorides[12–14]. The potential observed with aliphatic sulfonyl fluorides, in particular the peptide-type inhibitors[12,22–26] (Fig. 1a), has drawn our attention from a synthetic point of view. Typically, aliphatic sulfonyl fluorides are prepared via fluoride-chloride exchange with the corresponding sulfonyl chlorides, which can be prepared from thiols, halides, or sultones (Fig. 1b). However, these methods have very limited sources of starting compounds[2]. Accordingly, a general and facile synthetic route to expand the toolkit of sulfonyl fluorides on both dimensions of synthetic efficiency and structural diversity is in high demand[12–14].

Carboxylic acid is ubiquitous in nature, and widely present in natural products, medicines, and materials. The merits such as stability, low toxicity, and commercial availability make carboxylic acids as an ideal type of building blocks in organic synthesis[27]. We conceived that the combination of carboxylic acid as a radical source with an appropriate $SO_2F$-containing radical acceptor would forge a general approach to various sulfonyl fluorides (Fig. 1c). With this aim in mind, we found carboxylic acids are known to readily undergo decarboxylation to release

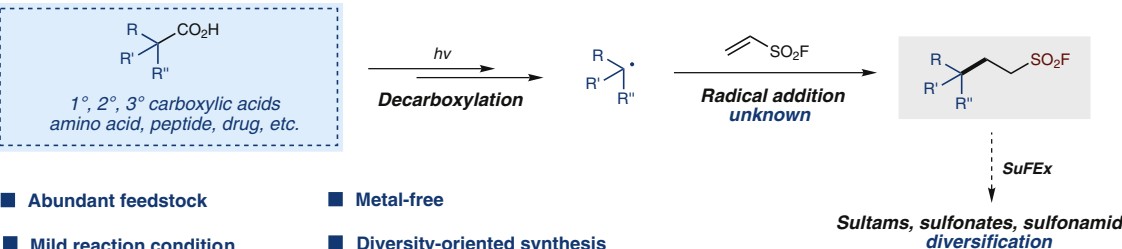

**Fig. 1** Aliphatic $RSO_2F$ inhibitors and the synthetic strategies. **a** Selected examples of aliphatic sulfonyl fluoride inhibitors. **b** Conventional synthetic routes to aliphatic sulfonyl fluorides. **c** Synthesis of aliphatic sulfonyl fluorides from carboxylic acids via visible-light-mediated decarboxylation (this work)

**Table 1 Reaction optimization (Reaction conditions: 0.05 mmol scale, VSF (2 eq), photocatalyst (5 mol%), in MeCN (0.5 mL) at room temperature, under the irradiation of 18 W × 2 blue LED bulbs. Left entries: with DIPEA (2 eq) as the reductant. Right entries: with Eosin Y-Na$_2$ (5 mol%) as the catalyst)**

| Entry | Photocatalyst | Yield[a] | Entry | Reductant | Yield[a] |
|---|---|---|---|---|---|
| 1 | Ir(ppy)$_3$ | 29% | 10 | TEA | <2% |
| 2 | [Ru(bpy)$_3$]Cl$_2$·6H$_2$O | 24% | 11 | DMA | N.P. |
| 3 | Eosin Y | 35% | 12 | TMEDA | <2% |
| 4 | Fluorescein | 26% | 13 | PMDETA | N.P. |
| 5 | Rhodamine B | 17% | 14 | HE | 89% |
| 6 | Rhodamine 6G | 26% | 15 | HE/DIPEA 2:1 | 73% |
| 7 | Isatin | 32% | 16 | HE/TEA 2:1 | 29% |
| 8[b] | Eosin Y | 38% | 17[c] | HE | 69% |
| 9 | Eosin Y-Na$_2$ | 37% | 18[d] | HE | N.P. |

*DMA N,N'-dimethylaniline, TMEDA N,N,N',N'-tetramethylethanediamine, PMDETA N,N,N',N'',N''-pentamethyldiethylenetriamine*
[a]Determined by $^{19}$F NMR with PhCF$_3$ as an internal standard or by GC-MS analysis
[b]Eosin Y (10 mol%), VSF (5 eq), in DCM, green LED, 18 h
[c]With 1 eq of VSF
[d]In dark, N.P. = no product was observed

alkyl radicals after activated in the form of N-hydroxyphthalimide (NHPI) esters[28–30] via photocatalysis[31–43] or low-valent transition metal catalysis[44–48]. While vinyl sulfonyl fluoride (VSF), a readily available SO$_2$F-containing reagent[49–54], could potentially be employed as the radical acceptor, though its utilization and compatibility in the radical reactions is unknown. Herein, we report our effort on this approach and the invention of a facile and general radical method for the synthesis of aliphatic sulfonyl fluorides based on naturally abundant carboxylic acid feed stock. This visible-light-mediated transformation features its high efficiency and a broad reaction scope. The wide availability of carboxylic acids, including amino acids, peptides, and many pharmaceuticals, allows for a fast construction of aliphatic sulfonyl fluoride libraries of considerable structural diversity. Further diversification of the SO$_2$F-containing products is also demonstrated in the derivatization to pharmaceutically important motifs such as sultams, sulfonates, and sulfonamides.

## Results

**Reaction optimization.** We commenced our study by using dihydrocinnamic acid-derived redox active ester (**1**) as the model substrate and diisopropylethylamine (DIPEA) as a reductant for photocatalyst screening. As shown in Table 1, with the commonly used transition metal photocatalysts, *fac*-Ir(ppy)$_3$ gave the desired product in 29% yield (entry 1), while lower yield (24%) was obtained with [Ru(bpy)$_3$]Cl$_2$·6H$_2$O (entry 2). To our delight, eosin Y, a cheap and metal-free organic dye could furnish a slightly better yield (35%, entry 3). Importantly, this metal-free system could avoid the transition metal contaminant which may bring about detrimental effect in biological and related applications[55]. Encouraged by this result, more organic dyes were screened (entries 4–7), such as fluorescein, rhodamine B, rhodamine 6G, riboflavin and isatin, but none of them could give a better yield. Further increasing the VSF loading to five equivalents and the catalyst loading to 10 mol% could slightly improve the yield from 35 to 38%, but the improvement was quite limited

(entry 8 vs. entry 3). Nevertheless, the disodium salt of eosin Y was identified slightly more efficient than eosin Y, giving 37% yield with two equivalents of VSF only and a lower catalyst loading (5 mol%, entry 9).

Vinyl sulfonyl fluoride (VSF) is known about six orders of magnitude more reactive than vinyl phenyl sulfone as a Michael acceptor[51]; however, under this photoredox catalytic reaction condition, the yield (37%) was even lower than that of vinyl phenyl sulfone (45%), showing no correlation to the acceptor activity in this reaction. To our surprise, following the reaction by $^{19}$F NMR unveiled a range of mass peaks between 50 to 60 ppm, which suggested severe side reactions happened, probably due to the high reactivity of VSF and its poor compatibility with the photoredox conditions. Further reaction optimization (reductant amines, solvent, light source, etc.; see Supplementary Table 1–3) was thus performed. Surprisingly, when DIPEA was replaced with triethylamine (TEA), almost no product formed (entry 10). This dramatic effect of amine reductant was unexpected, which urged us to follow the reaction by NMR analysis and also carry out several control experiments (see Supplementary Discussion). In both cases of DIPEA and TEA, the redox active ester **1** were completely consumed after 12 h, and almost no VSF left, so the low yields did not result from low conversion. Notably, for control reactions without redox active ester **1**, the $^{19}$F NMR also showed the disappearance of the peak of VSF, indicating that VSF may be unstable and prone to polymerize[56] in the presence of DIPEA or TEA under the photoredox conditions. Therefore, more amines were tested, and finally, Hantzsch ester (HE, diethyl 1,4-dihydro-2,6-dimethyl-3,5-pyridinedicarboxylate) was found to be uniquely beneficial to the reaction, able to shut down the undesired reaction pathways and afford **3** in a decent yield of 89% (entry 14), probably by a fast hydrogen atom transfer process to the radical adduct (for mechanistic study, isotope-labeling experiments and a proposed mechanism, please see Supplementary Discussion and Supplementary Figs. 3–5). NMR experiments showed that VSF was much more stable in the presence of HE,

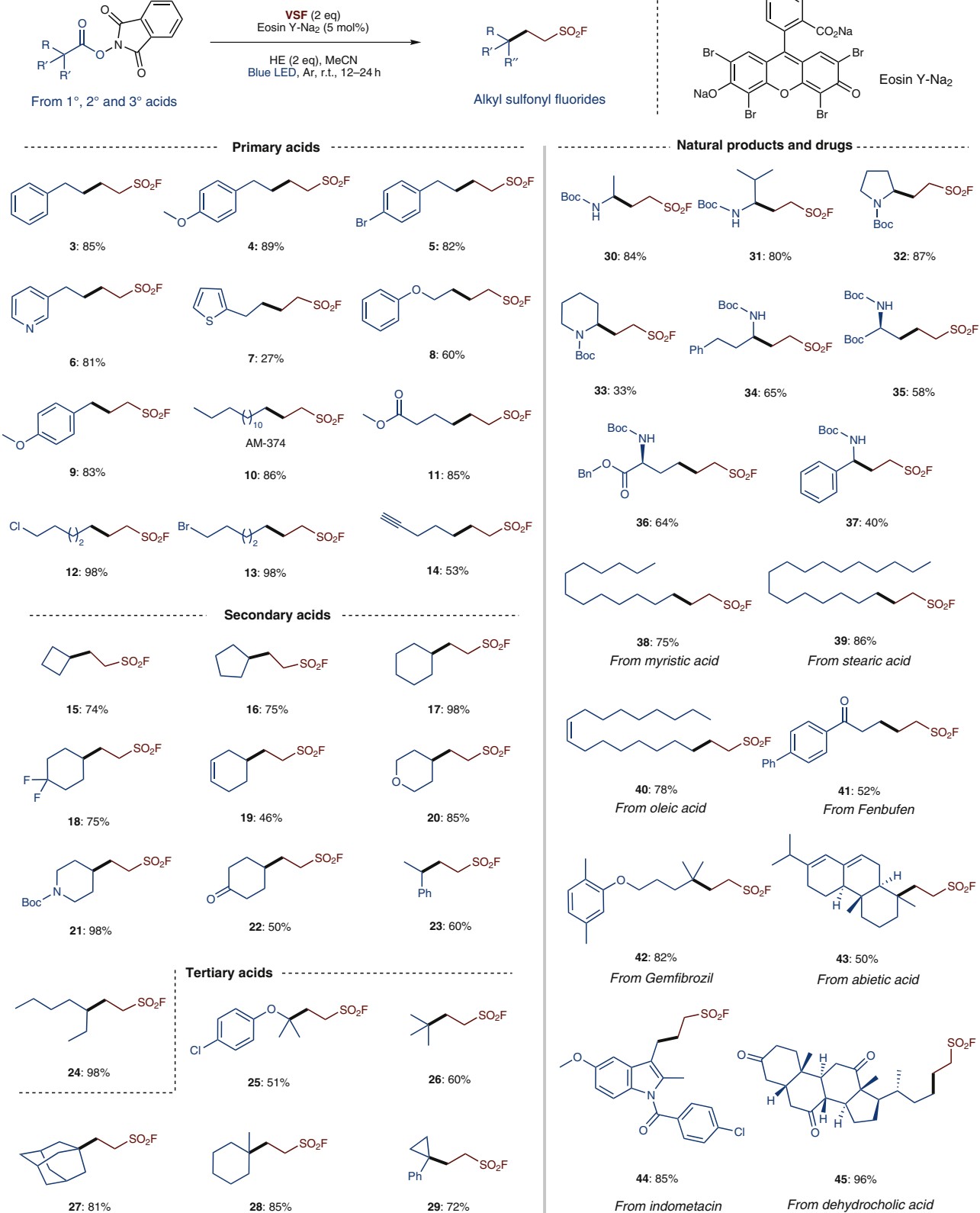

**Fig. 2** Substrate scope. Reactions were performed on a 0.2 mmol scale, and all yields represent isolated yields. Substrates for **33**, **34** and **37** are unnaturally occurring amino acids

and even till the end of the reaction, more than 90% of the excessive one equivalent of VSF remained intact. Moreover, mixing VSF and HE without **1** under the same irradiation conditions for 12 h showed no detectable loss of VSF, in sharp contrast to the cases of DIPEA and TEA (for comparison, see Supplementary Figs. 1 and 2). The addition of DIPEA or TEA was even detrimental to the reaction, leading to a dramatic drop in yield (entries 15 and 16); further confirming this catalytic system

(Eosin Y-Na$_2$/HE) is superior to the known Eosin Y/DIPEA combination to ensure a high reaction yield. It is worth mentioning that the product could be obtained in 69% yield with one equivalent of VSF only (entry 17), while without light no reaction was observed (entry 18).

**Substrate scope**. With the catalytic synthetic method (Eosin Y-Na$_2$/HE) successfully established, we next examined the reaction scope with a variety of primary, secondary and tertiary acids (Fig. 2). In cases of primary acids, bromide (**5** and **13**) and chloride substituents (**12**) were well tolerated under the present photoredox conditions and excellent yields can be achieved. Acids containing a heteroarene group like pyridine (**6**) or thiophene (**7**)

are suitable substrates, though lower yield was obtained in the latter case. Acid with an alkyne moiety could deliver the desired product (**14**) in a moderate yield. Secondary acids with different ring sizes, all reacted well and high yields were obtained (**15–22**). Acyclic ones were also transformed readily (**23** and **24**). Tertiary acids such as alpha-oxo-acid and simple pivalic acid delivered the products **25** and **26** in good yields, respectively. Reactions could also proceed well on the six- and three-membered ring site (**28** and **29**). To further examine the reaction scope, a number of naturally occurring carboxylic acids and several drugs were tested. High yields were in general achieved. The decarboxylative fluorosulfonylethylation of amino acids (**30–34**) allows the introduction of an SO$_2$F group at the C-terminus of an amino acid chain, while the side-chain carboxylic acid group of aspartic

**Fig. 3** Reaction practicability and diversification of products. **a** Scale-up, one-pot reaction. **b** Modification of peptide. **c** Diversification of products through SuFEx chemistry

acid and glutamic acid could also be utilized (**35** and **36**). The low isolated yield of product **33** is due to the difficult separation from the pyridine-byproduct of Hantzsch ester by chromatography, as the $^{19}$F NMR yield is over 90%. Furthermore, the carboxylic groups of drugs such as Fenbufen, Gemfibrozil, and Indometacin could be modified readily in good to high yields to introduce the sulfonyl fluoride function through this decarboxylative approach (**41–42** and **44**). As outlined in Fig. 2, under the current photoredox catalytic system, primary, secondary, and tertiary carbon radicals can all be readily generated from the corresponding redox active esters and trapped by VSF in high efficiency, allowing for a fast construction of aliphatic sulfonyl fluoride libraries of considerable structural diversity. And noteworthy, primary (**9**), secondary (**23** and **37**), and tertiary (**29**) benzylic radicals are also well accommodated.

**Product diversification**. Regarding reaction practicality, the free acid can be directly used for this reaction by forming the redox active ester in situ, as exemplified in a scaled-up reaction with *N*-Boc-protected L-proline (Fig. 3a). The in situ procedure proved to be equally efficient and afforded the desired product **32** in 85% isolated yield. This convenient, one-pot procedure was later employed to modify the *C*-end of a tetrapeptide (**46**) (Fig. 3b). The SO$_2$F-containing product, peptide **47** is a homolog of **PW28** (Fig. 1a) which exhibited high inhibition activity to proteasome (IC$_{50}$ 7 nm)[22,23]. In addition, we tested the stability of the obtained product **3**, **32** and tetrapeptide **47** in physiological buffers (PBS, pH 7.2), which unveiled that the three sulfonyl fluorides are quite stable under this aqueous conditions, without detectable loss or decomposition after 24 h by $^{19}$F NMR analysis (see Supplementary Note 1 and Supplementary Figs. 6–8). Moreover, the γ-amino sulfonyl fluoride products such as **30–34** derived from the α-amino acids are new compounds and to some extent can be regarded as a type of activated γ-amino sulfonic acids, which would be expected to undergo cyclization or ligation to other molecules to construct pharmaceutically important motifs such as sultam, sulfonate, sulfonamide etc.[57–61].

As demonstrated in Fig. 3c, bicyclic sultams **48** and **49** could be obtained in high yields via a deprotection-intramolecular cyclization sequence, which represents a facile synthetic route to sultams[58]. On the other hand, proline-derived sulfonyl fluoride **32** could be further diversified through SuFEx click reactions with phenol, morpholine, and TMSN$_3$, delivering the corresponding sulfonate **50**, sulfonamide **51**, and sulfonyl azide **52**, respectively, in high yields. Notably, the introduction of an azide group (**52**) allows for a further ligation to alkynes through azide-alkyne click reactions, as exemplified by the synthesis of triazole compound **53**.

## Discussion

In conclusion, a facile catalytic system has been successfully developed for the synthesis of aliphatic sulfonyl fluorides based on abundant carboxylic acid feed stock. A variety of alkyl carboxylic acids including primary, secondary, and tertiary ones, as well as amino acids, peptides, and several drugs can be readily transformed into the corresponding alkyl sulfonyl fluorides with high reaction efficiency and structural diversity. Further diversification of the sulfonyl fluoride products has been also demonstrated through the intramolecular cyclizations and SuFEx reactions. The current method provides a general and rapid approach for the synthesis of aliphatic sulfonyl fluorides and derivatives, which expands the chemical biology toolkit and could benefit the related studies in the context of biology and drug discovery.

## Methods

**General procedure**. Under argon, to an oven-dried Schlenk tube (10 mL) equipped with a stir bar, was added NHPI redox active ester (0.2 mmol, 1 equiv.), Eosin Y-Na$_2$ (6.8 mg, 0.01 mmol, 0.05 equiv.), and HE (101.2 mg, 0.4 mmol, 2 equiv.), followed by the addition of dry MeCN (2 mL) and VSF (32 μL, 0.4 mmol, 2 equiv.). The reaction mixture was then degassed by three freeze-pump-thaw cycles. The Schlenk tube was then backfilled with argon. The reaction mixture was stirred at room temperature for 12−24 h under the irradiation of blue LED bulb (18 W × 2, at approximately 2 cm away from the light sources, ca. 25 °C). The product was purified by flash chromatography (SiO$_2$, petroleum ether/ethyl acetate = 20:1 to 4:1) to give the corresponding pure product. Full experimental details and characterization of new compounds can be found in the Supplementary Methods and Supplementary Figs. 9–108.

## Data availability

The authors declare that all data supporting the findings of this study are available within the article and Supplementary Information files, and are also available from the corresponding author upon reasonable request.

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

## Acknowledgements

We gratefully acknowledge the Recruitment Program of Global Experts (1000plan), Natural Science Foundation of China (Nr. 21602028), and Fuzhou University for the financial support.

## Author contributions

R.X. developed the reactions, and performed the reaction scope investigation, mechanistic study, and product derivatization. T.X. and T.C. participated in the synthesis of substrates, M.Y. helped the study of the reaction scope. S.L. conceived this concept and prepared this manuscript with feedback from R.X.

## Additional information

**Competing interests:** The authors declare no competing interests.

