## [Peer Review File · Nature Communications]

REVIEWERS' COMMENTS:

Reviewer #1 (Remarks to the Author):

To Author:

Thanks for submit 'A rapid access to aliphatic sulfonyl fluorides' to Nature Communication for reviewing. The authors demonstrated a novel method of light-catalyzed decarboxylative coupling reaction of alkyl NHPI ester with ethenesulfonyl fluoride to access aliphatic sulfonyl fluoride, which is widely used as synthetic synthons in SuFEx click chemistry and regarded as chemical probes in chemical biology field. The optimization of the methods was focused on choice of photo-catalyst. Authors has finally reached high yield with the right choice of base, Hantzsch ester, which is mild enough to avoid the decomposition of the final product sulfonyl fluoride. Due to the mild condition, substrate of scope is wide with primary, secondary, tertiary acid and natural product derivatives. The authors also demonstrated the synthetic application of aliphatic sulfonyl fluorides. Overall, this manuscript has a complete story and solid experiment data. It represents a novel method to synthesize aliphatic sulfonyl fluoride. I suggest this manuscript to be accept with minor revision. Here are the discussions:

1. From synthetic strategy wise, this method belongs to the photo-catalyzed cross-coupling of NHPI esters with nucleophile with free-radical mechanism. This field is very hot now and a lot of good papers was published recently. Authors should carefully choose reference to cite. For example, prof. Yao Fu has published a nickel-catalyzed decarboxylative hydroalkylation NHPI esters with olefins on Chem. Eur. J. 2016, 22, p11161, which represents an earlier example of hydroalkylation of olefins than authors.
2. Even though the manuscript is titled with 'access to aliphatic sulfonyl fluoride'. The final product inherited the sulfonyl fluoride structural motif from ethenesulfonyl fluoride (as VSF in the manuscript). As a famous Michael acceptor, VSF was functionalized via nucleophilic addition. This manuscript presumably represents a new strategy of applying free radical addition to VSF. Authors also cited several latest references about application of VSF. However, Donot forget citing the original John Hyatt's JOC paper (J. Org. Chem., 1979, 44, 3847).
3. The free radical mechanism of light-catalyzed NHPI ester coupling reaction has been well studied and documented. Authors also proposed a mechanism in Supporting Information. For the general scientific readers' view, I suggest the authors to move the mechanism scheme to the main context.
4. The data format in supporting information need careful check. ¹³C NMR data normally only record 1 digit after '.', such as 169.2, instead of 169.23.

Reviewer #2 (Remarks to the Author):

This manuscript submitted by Liao and coworkers describes a novel and general approach for the synthesis of aliphatic sulfonyl fluorides via visible-light mediated decarboxylative fluorosulfonylethylation. Aliphatic sulfonyl fluorides are not that easy to make comparing with known SuFEx hubs, and their claims made for the importance of such type of compounds are correct in my opinion. Carboxylic acids are very common and versatile. I suspect many chemical biology labs would be able to take advantage of this method for the synthesis of their probes. I am therefore supportive of publishing this paper in Nature communication after minor revisions.

1) The scholarly presentation of these results is excellent, and I saw no confused presentations in their draft. The reaction condition seems simple and it is not too hard for other labs to repeat their procedure.

2) Based on the reported data, the substrate scope of this reaction is good.

3) Although there were reported procedures using visible-light-mediated decarboxylation and Michael acceptor before. However, I suspect this procedure holds unique potential for utility discovery. Light-induced radical reactions are not easy to scale up, but it is also true that mg scale chemical probes are well enough for chemical biology experiments in many labs. Moreover, this method provides huge chemical space for aliphatic sulfonyl fluoride probes, and provide access to experiments which otherwise would likely never have been tried.

4) The paper's conclusions are correct by the data presented.

5) I have little to offer for suggested improvements, but I know different types of S(VI)-F functional groups would have different stability in buffers. For example, PMSF's half-life is very short in physiological buffers (~30mins), but the sulfonyl fluorides presented in this manuscript could be much more stable. It will be useful to provide some simple experiments to demonstrate the unique properties of this type of probes for many readers who are not aware of the potential applications.

Dear Editors and Reviewers,

First we would like to thank the Reviewers for the comments and suggestion. We have revised the manuscript and supplementary information accordingly. Below are our point-by-point responses:

Reviewer #1:

Thanks for submit 'A rapid access to aliphatic sulfonyl fluorides' to Nature Communication for reviewing. The authors demonstrated a novel method of light-catalyzed decarboxylative coupling reaction of alkyl NHPI ester with ethenesulfonyl fluoride to access aliphatic sulfonyl fluoride, which is widely used as synthetic synthons in SuFEx click chemistry and regarded as chemical probes in chemical biology field. The optimization of the methods was focused on choice of photo-catalyst. Authors has finally reached high yield with the right choice of base, Hantzsch ester, which is mild enough to avoid the decomposition of the final product sulfonyl fluoride. Due to the mild condition, substrate of scope is wide with primary, secondary, tertiary acid and natural product derivatives. The authors also demonstrated the synthetic application of aliphatic sulfonyl fluorides. Overall, this manuscript has a complete story and solid experiment data. It represents a novel method to synthesize aliphatic sulfonyl fluoride. I suggest this manuscript to be accept with minor revision. Here are the discussions:

1. From synthetic strategy wise, this method belongs to the photo-catalyzed cross-coupling of NHPI esters with nucleophile with free-radical mechanism. This field is very hot now and a lot of good papers was published recently. Authors should carefully choose reference to cite. For example, prof. Yao Fu has published a nickel-catalyzed decarboxylative hydroalkylation NHPI esters with olefins on Chem. Eur. J. 2016, 22, p11161, which represents an earlier example of hydroalkylation of olefins than authors.

Response: Thank you very much for the comments and suggestion. As suggested, we have cited this paper as Ref. 44 in the revised manuscript and also added three newly published and related papers as Ref. 37-39 (Nat. Commun. 2018, 9, 1; Angew. Chem. Int. Ed. 2019, 58, 10514; Science 2019, 363, 1429).

2. Even though the manuscript is titled with 'access to aliphatic sulfonyl fluoride'. The final product inherited the sulfonyl fluoride structural motif from ethenesulfonyl fluoride (as VSF in the manuscript). As a famous Michael acceptor, VSF was functionalized via nucleophilic addition. This manuscript presumably represents a new strategy of applying free radical addition to VSF. Authors also cited several latest references about application of VSF. However, Donot forget citing the original John Hyatt's JOC paper (J. Org. Chem., 1979, 44, 3847).

Response: We have cited this important paper by John Hyatt as Ref. 48 in the revised manuscript. Thank you very much for the suggestion.

3. The free radical mechanism of light-catalyzed NHPI ester coupling reaction has been well studied and documented. Authors also proposed a mechanism in Supporting Information. For the general scientific readers' view, I suggest the authors to move the mechanism scheme to the main context.

Response: Thank you very much for the comments and suggestion. We tried to move the proposed mechanism to the main context and give some description, but we found it exceeded the length limit. Thus, we added “for mechanistic study, isotope-labeling experiment and a proposed mechanism, please see Supplementary Discussion and Supplementary Figs. 3-5” in the main text of the revised manuscript (Page 4, Line 27) for readers’ interest and also the convenience to locate the mechanistic part. We hope this revision is fine. Thank you again for the suggestion.

4. The data format in supporting information need careful check. ^{13}C NMR data normally only record 1 digit after ‘.’, such as 169.2, instead of 169.23.

Response: Thank you very much for pointing out this important issue. We have corrected the format of all the ^{13}C and also ^{19}F NMR data

Reviewer #2:

This manuscript submitted by Liao and coworkers describes a novel and general approach for the synthesis of aliphatic sulfonyl fluorides via visible-light mediated decarboxylative fluorosulfonylation. Aliphatic sulfonyl fluorides are not that easy to make comparing with known SuFEx hubs, and their claims made for the importance of such type of compounds are correct in my opinion. Carboxylic acids are very common and versatile. I suspect many chemical biology labs would be able to take advantage of this method for the synthesis of their probes. I am therefore supportive of publishing this paper in Nature communication after minor revisions.

1) The scholarly presentation of these results is excellent, and I saw no confused presentations in their draft. The reaction condition seems simple and It is not too hard for other labs to repeat their procedure.

Response: Thank you very much for the comments.

2) Based on the reported data, the substrate scope of this reaction is good.

Response: Thank you very much for the comments.

3) Although there were reported procedures using visible-light-mediated decarboxylation and Michael acceptor before. However, I suspect this procedure holds unique potential for utility discovery. Light-induced radical reactions are not easy to scale up, but it is also true that mg scale chemical probes are well enough for chemical biology experiments in many labs. Moreover, this method provides huge chemical space for aliphatic sulfonyl fluoride probes, and provide access to experiments which otherwise would likely never have been tried.

Response: Thank you very much for the comments and pointing out the significance of this synthetic strategy.

4) The paper's conclusions are correct by the data presented.

Response: Thank you very much for the comments.

5) I have little to offer for suggested improvements, but I know different types of S(VI)-F functional groups would have different stability in buffers. For example, PMSF's half-life is very short in physiological buffers (~30mins), but the sulfonyl fluorides presented in this manuscript could be much more stable. It will be useful to provide some simple experiments to demonstrate the unique properties of this type of probes for many readers who are not aware of the potential applications.

Response: Thank you very much for the comments and suggestion. The stability of the sulfonyl fluorides is an important issue for related biological investigation.

As suggested, we performed the stability study with product 3, 32, and also the tetrapeptide 47 under a physiological buffer (PBS, pH 7.2), with $\text{CF}_3\text{CH}_2\text{OH}$ as an internal standard for ^{19}F NMR monitoring. We checked the NMR after 2h, 4h, 8h, 24h, which indicated that almost no detectable loss or decomposition of the three compound in this physiological buffer solution by ^{19}F NMR analysis. We have added "In addition, we tested the stability of the obtained product 3, 32 and tetrapeptide 47 in physiological buffers (PBS, pH 7.2), which unveiled that the three sulfonyl fluorides are quite stable under this aqueous conditions, without detectable loss or decomposition after 24 h by ^{19}F NMR analysis (see Supplementary Note 1)" in the revised manuscript, and the detailed study is described as Supplementary Note 1 in the Supplementary Information.